# Faraday Rotation of Dy_2_O_3_, CeF_3_ and Y_3_Fe_5_O_12_ at the Mid-Infrared Wavelengths

**DOI:** 10.3390/ma13235324

**Published:** 2020-11-24

**Authors:** David Vojna, Ondřej Slezák, Ryo Yasuhara, Hiroaki Furuse, Antonio Lucianetti, Tomáš Mocek

**Affiliations:** 1HiLASE Centre, FZU–Institute of Physics of the Czech Academy of Sciences, Za Radnicí 828, 252 41 Dolní Břežany, Czech Republic; slezako@fzu.cz (O.S.); lucianetti@fzu.cz (A.L.); mocek@fzu.cz (T.M.); 2Department of Physical Electronics, Faculty of Nuclear Sciences and Physical Engineering, Czech Technical University in Prague, Břehová 7, 115 19 Prague, Czech Republic; 3National Institute for Fusion Science, National Institutes of Natural Sciences, 322-6, Oroshi-cho, Toki, Gifu 509-5292, Japan; yasuhara@nifs.ac.jp; 4Kitami Institute of Technology, 165 Koen-cho, Kitami, Hokkaido 090-8507, Japan; furuse@mail.kitami-it.ac.jp

**Keywords:** Faraday effect, magneto-optical properties, Verdet constant, Faraday rotation, magneto-active materials, yttrium iron garnet, dysprosium sesquioxide, cerium fluoride, Faraday devices, mid-infrared lasers

## Abstract

The relatively narrow choice of magneto-active materials that could be used to construct Faraday devices (such as rotators or isolators) for the mid-infrared wavelengths arguably represents a pressing issue that is currently limiting the development of the mid-infrared lasers. Furthermore, the knowledge of the magneto-optical properties of the yet-reported mid-infrared magneto-active materials is usually restricted to a single wavelength only. To address this issue, we have dedicated this work to a comprehensive investigation of the magneto-optical properties of both the emerging (Dy${}_{2}$O${}_{3}$ ceramics and CeF${}_{3}$ crystal) and established (Y${}_{3}$Fe${}_{5}$O${}_{12}$ crystal) mid-infrared magneto-active materials. A broadband radiation source was used in a combination with an advanced polarization-stepping method, enabling an in-depth analysis of the wavelength dependence of the investigated materials’ Faraday rotation. We were able to derive approximate models for the examined dependence, which, as we believe, may be conveniently used for designing the needed mid-infrared Faraday devices for lasers with the emission wavelengths in the 2-μm spectral region. In the case of Y${}_{3}$Fe${}_{5}$O${}_{12}$ crystal, the derived model may be used as a rough approximation of the material’s saturated Faraday rotation even beyond the 2-μm wavelengths.

## 1. Introduction

A considerable amount of attention from the scientific community is currently directed towards the development of the mid-infrared (mid-IR) lasers. The main reason is that these laser sources possess the key properties for a vast number of technological applications (e.g., LiDAR, gas sensing, material processing, or pump sources of optical parametric oscillators [1,2,3]) and are enabling further advancement of the basic research as well (e.g., in the attosecond science for efficient high harmonic X-ray generation [4,5,6]). The huge demand for the mid-IR lasers has inevitably led to an intensified development of the corresponding laser sources and optical components [1,7,8,9,10,11,12].

One of the most commonly used components in laser systems are those based on the Faraday effect, incorporating, for instance, magneto-optical modulators, polarization switches, rotators, deflectors, or isolators. All of these are frequently used for the polarization-based signal processing of the laser beam, construction of multi-pass amplifiers, or as a protection of the laser system from back-reflections [13,14,15]. The fundamental part of these devices is a piece of magneto-active material in which the plane of the polarized radiation is rotated by a certain angle as a result of applying the magnetic field.

A great portion of research dedicated to the development of Faraday devices now focuses on the investigation of new magneto-active materials, which would enable running these devices with high efficiency. The basic requirements on an efficient magneto-active material are to exhibit a high Faraday rotation, which is represented by the Verdet constant or the specific Faraday rotation parameters, and low absorption at the incident laser wavelength. In the case of a Faraday device designed for a high average power laser, the list of the material parameters which needs to be further assessed incorporates the thermo-optical, thermo-mechanical and elasto-optical properties of the material [16,17,18,19].

While a lot of scientific effort has been already put into the investigation of the magneto-active materials for the visible and near-IR wavelengths, the options for the mid-IR wavelengths are still very limited [18]. The materials used for the near-IR wavelengths usually include Tb${}^{3+}$ ions (e.g., the terbium-based garnets, fluorides or sesquioxides have been reported [20,21,22,23]), which have strong absorption in the mid-IR and, therefore, cannot be used. For low-power mid-IR laser applications, the ferrimagnetic yttrium iron garnet Y${}_{3}$Fe${}_{5}$O${}_{12}$ (YIG) crystal and its rare-earth-substituted compositions may be used, as these possess a very high specific Faraday rotation and low saturation magnetization [13,24]. Nevertheless, the fabrication of these crystals is a relatively time-consuming process due to the need for using the floating zone or liquid phase epitaxial methods. These drawbacks have recently prompted researchers to explore the possibility of fabricating the iron garnets in the form of transparent ceramics which would enable reducing the time needed for obtaining the iron-garnet-based magneto-optical elements as well as to manufacture them in larger sizes [25,26,27].

Most recently, quite a few paramagnetic or diamagnetic mid-IR magneto-active materials have been reported as well. Unlike to the ferrimagnetic iron garnets, the saturation of the Faraday effect is much less pronounced in these materials and, hence, they may be used in the linear regime of operation in which the polarization plane rotation angle is tuned by adjusting the applied magnetic field rather than by adjusting the length of the magneto-optical element. Concerning the paramagnetic materials, favourable optical characteristics for the mid-IR region were reported for fluorides including rare-earth ions [28], specifically for the CeF${}_{3}$, PrF${}_{3}$ and LiREF${}_{4}$ (RE = Dy, Ho or Er). Among these, to the best of our knowledge, only the properties of the CeF${}_{3}$ crystal were further investigated in the context of the Faraday devices [29,30,31,32]; and, concerning the mid-IR, its Verdet constant is still known only up to 1.95μm [31]. Very recently, the mid-IR magneto-optical characteristics were reported for the (Dy${}_{X}$Y${}_{0.95-X}$La${}_{0.05}$)${}_{2}$O${}_{3}$ ceramics [33] (with variable doping X = 0.7, 0.8 and 0.9), EuF${}_{2}$ crystal [34], Dy-doped multicomponent glasses [35] and for a tellurium-arsenic-selenium glass [36], all of them showing a great promise for the mid-IR region. In these papers, the room temperature Verdet constant on a single wavelength of ∼1.94μm was reported for all of the mentioned materials to be in the range of ∼8–15 rad/Tm in the absolute value (see Table 1). After the needed optimization of the fabrication process and further characterization, these materials may be used for the mid-IR Faraday devices, nevertheless, given the reported values of the Verdet constant, their use might require relatively strong magnetic fields >2T. Permanent magnets with such peak magnetic field magnitudes were already reported [37,38,39]; however, it might be quite challenging to achieve the desired axial magnetic field over the whole needed length of the magneto-optical element. Therefore, searching for a non-ferrimagnetic mid-IR magneto-active material with higher magneto-optical characteristics still represents an up-to-date task.

In our recent work [40], we have investigated the Verdet constant of a pure Dy${}_{2}$O${}_{3}$ ceramics as a function of wavelength and temperature. Compared to the Dy-Y-La-based ceramics [33], the pure Dy${}_{2}$O${}_{3}$ has a larger concentration of the highly magnetically active Dy${}^{3+}$ ions, and, thereby, a larger magneto-optical activity was observed at the visible and near-IR wavelengths. In the study, we have specifically demonstrated the importance of consideration of the Dy${}^{3+}$ ion’s multiple transitions contribution to the Verdet constant of the Dy${}_{2}$O${}_{3}$ ceramics, nevertheless, the obtained data in the mid-IR suffered from a large measurement error due to the limited optical quality and short length of the characterized sample.


In this paper, we report on the fabrication of a new material sample of Dy${}_{2}$O${}_{3}$ ceramics and a comprehensive investigation of the magneto-optical properties with a focus on the mid-IR region. Apart from the Dy${}_{2}$O${}_{3}$ ceramics, we have also investigated samples of YIG and CeF${}_{3}$ crystals under the same experimental conditions. The knowledge of the magneto-optical properties of both of these materials is incomplete in their mid-IR transparency windows, which, although these materials are commercially available, is currently hampering their practical implementation in the mid-IR Faraday devices.

For all samples under investigation, we have measured the optical in-line transmittance (0.3–3μm) and the Verdet constant dispersion (0.6–2.3μm for the Dy${}_{2}$O${}_{3}$ ceramics and CeF${}_{3}$ crystal) or the specific Faraday rotation (1.1–2.3μm for the YIG crystal) at the room temperature. A broadband radiation source was used in our investigation in a combination with an advanced polarization-stepping method, enabling an in-depth analysis of the Faraday rotation dispersion of the investigated materials in the whole 2-μm region. The validity of the obtained data, errors of the measurement, and further prospects of the materials’ use in the mid-IR Faraday devices are discussed in detail.

## 2. Methods of investigation

### 2.1. Fabrication Process of Dy${}_{2}$O${}_{3}$ Ceramics and Sample Parameters

The sample of Dy${}_{2}$O${}_{3}$ ceramics investigated in this study was manufactured using spark plasma sintering technique at Kitami Institute of Technology, similarly as in the [41]. Commercially obtained Dy${}_{2}$O${}_{3}$ powder (Shin-Etsu Chemical Co., Tokyo, Japan; 99.9% purity) with an average particle size of 340 nm was used. The powder was put into a graphite die and sintered under vacuum with uniaxial pressing. The sintering temperature was 1100 ${}^{{}^{\circ}}C$; the applied pressure at the nominal temperature was equal to 80 MPa. After the sintering process, the sample was annealed in the air to compensate for the oxygen defects and to remove carbon contamination. Both surfaces of the annealed sample were polished for the characterization. The final form of the fabricated Dy${}_{2}$O${}_{3}$ ceramics sample is shown in the Figure 1, on the right, along with the commercially obtained CeF${}_{3}$ (Kingheng Crystal Co., Shanghai, China) and YIG (Oxide Co., Yamanashi, Japan) crystal samples. Thicknesses of the depicted samples were 0.192±0.001mm, 7.553±0.001mm and 2.006±0.001mm for the Dy${}_{2}$O${}_{3}$ ceramics, CeF${}_{3}$ crystal and YIG crystal samples, respectively. The diameter of the Dy${}_{2}$O${}_{3}$ ceramics and CeF${}_{3}$ crystal samples was 10mm; the YIG crystal sample had 5mm in diameter. No coatings were applied to any of the samples.

### 2.2. Optical Transmittance Measurement

The in-line optical transmittance of the investigated samples was measured in the 0.3–3μm spectral range using the Shimadzu UV-3600 Plus spectrophotometer. The obtained results are shown in the Figure 1, on the left. As may be observed, the fabricated Dy${}_{2}$O${}_{3}$ ceramics (red line) possesses several transparency windows/peaks at approximately: (a) 0.5–0.7μm, (b) 0.84μm, (c) 0.98μm, (d) 1.5μm, (e) 1.86–2.3μm and (f) 2.95μm. In contrast with the Dy${}_{2}$O${}_{3}$ ceramics, the CeF${}_{3}$ crystal (yellow line) has a wide window of transparency ranging from 0.3μm (the material’s cut-off wavelength is located at ∼0.24μm [28], which is not detectable with the spectrophotometer that was used) to 2.5μm. The YIG crystal (blue line) is transparent from 1.1μm to ∼5μm [13,42,43,44]. The transparency windows of these materials nicely match the emission wavelengths of some of the visible, near-IR and, most interestingly for our investigation, some of the Tm${}^{3+}$, Ho${}^{3+}$ or Er${}^{3+}$-doped lasers emitting around 2 and 3μm wavelengths mentioned in the introduction [1,8,9,45].

It needs to be noted that the fabrication process of the Dy${}_{2}$O${}_{3}$ ceramics has to be further optimized before the ceramics will be suitable for a practical application in the mid-IR Faraday devices. By a careful choice of the sintering conditions and additives, it will be eventually possible to increase the transmittance in the individual transparency regions. The first study of such a kind has been already published quite recently [46]. The authors of the paper have used ZrO${}_{2}$ powder as a sintering aid and have obtained samples with an enhanced optical quality compared to the one we have used in our investigation. Nevertheless, the fabricated ceramics’ magneto-optical properties were studied only on a single wavelength in the visible (on the 0.633μm) and, therefore, the knowledge of the Verdet constant of the Dy${}_{2}$O${}_{3}$ ceramics in the mid-IR region still remains incomplete due to the poor accuracy of the only available mid-IR data [40].

### 2.3. Faraday Rotation Dispersion Measurement

The method used for the Faraday rotation dispersion characterization is a modified version of the polarization-stepping technique. The method has been already described in a great level of detail [18,31,41] and, hence, we are going to describe only its basic principles within the scope of this study. A simplified scheme of the experimental setup is depicted in Figure 2. In the experiment, a probe beam from the broadband radiation source (NKT Photonics SuperK Compact, 0.4–2.3μm) is directed into a high-contrast Glan polarizer and then enters the investigated material sample, which is fixed in a special optomechanical holder (not visible in the scheme). If an external magnetic field is applied on the sample, the propagating probe beam’s polarization plane is being rotated because of the Faraday effect. The induced rotation is analysed in the detection system consisting of an achromatic half-wave plate (HWP)—Thorlabs SAHWP05M-1700 (0.6–2.7μm), which may be arbitrarily rotated around the optical axis by an angle α, an output Glan polarizer, and two spectrometers—Hamamatsu TG-C11118GA (0.9–2.55μm) and Ocean Optics HR4000CG-UV-NIR (0.2–1.1μm). The total effective spectral range of the experimental setup is 0.6–2.3μm.

The measurement is taken twice in the experiment—with and without applying the magnetic field. The spectra gathered in each of the measurements are taken in a large number of angular steps of the HWP (a quarter-circle total rotation is done with a 0.2 degree angular step). The data gathered in the individual measurements is proportional to the following cosine-squared function
(1)$$I_{n}\left(\alpha ,\lambda_{i}\right)=\cos^{2}\;\left[2\alpha +\alpha_{0}\left(\lambda_{i}\right)\right],$$
where the $I_{n}\left(\alpha ,\lambda_{i}\right)$ denotes the measured intensity data normalized to a unit interval $\langle 0,1\rangle$ for each detectable wavelength $\lambda_{i}$, α is the HWP’s angular position and $\alpha_{0}\left(\lambda_{i}\right)$ denotes the fitting parameter, the initial phase shift. The induced polarization rotation angle $\theta (\lambda_{i})$ is calculated by subtracting the initial phase shift obtained for the measurement with the magnetic field from the value obtained for the measurement without it. In the latter case, the initial phase shift is proportional to the mutual angular position of the polarizers and to the initial HWP fast-axis bias angle.

In our investigations, we have also performed the Faraday rotation measurement of the Dy${}_{2}$O${}_{3}$ and CeF${}_{3}$ samples using a He-Ne laser at 0.633μm wavelength. Its purpose is to have another reference measurement, which could support the validity of the broadband measurement. For the case of laser measurement, the detection system is modified by exchanging the spectrometers for a power meter. The experimental procedure is similar to the one described above.

### 2.4. Faraday Rotation in the Linear and in the Saturated Regime of the Faraday Effect

The Faraday rotation dispersion in the linear (far from saturation) regime of the Faraday effect is given by
(2)$$\theta^{\left(l\right)}\left(\lambda \right)=\Phi^{\left(l\right)}\left(\lambda \right)L=V\left(\lambda \right)B_{\mathrm{eff}}L,$$
where $\Phi^{\left(l\right)}\left(\lambda \right)$ refers to the specific Faraday rotation in the linear regime, *L* denotes the length of the sample, $B_{\mathrm{eff}}$ is the effective value of the applied magnetic field and V(λ) is the Verdet constant dispersion. In the expression (Equation 2), the Verdet constant represent the material-dependent quantity which is used to describe the magnitude of the Faraday rotation induced by a material in the linear regime. We have investigated the CeF${}_{3}$ and Dy${}_{2}$O${}_{3}$ material samples in the linear regime as such an arrangement corresponds to the usual, expected regime of operation of the paramagnetic materials in Faraday devices.

Unlike to the Dy${}_{2}$O${}_{3}$ and CeF${}_{3}$, the YIG crystal sample was characterized in the saturated regime of the Faraday effect, in which the polarization rotation angle induced by the material does not grow further with an increasing strength of the applied magnetic field. This arrangement, again, corresponds to the usual regime of operation of the ferrimagnetic materials in the Faraday devices. The Faraday rotation dispersion in the saturated regime is given by
(3)$$\theta^{\left(s\right)}\left(\lambda \right)=\Phi^{\left(s\right)}\left(\lambda \right)L,$$
where $\Phi^{\left(s\right)}\left(\lambda \right)$ is the specific Faraday rotation for the saturated regime, which, in this case, represents the material-dependent quantity that describes the magnitude of the Faraday rotation induced by the material.

The values of *L* and $B_{\mathrm{eff}}$, which were eventually used in the calculations of $V(\lambda_{i})$ and $\Phi^{\left(s\right)}\left(\lambda_{i}\right)$ at the detectable wavelengths $\lambda_{i}$ using the measured polarization rotation angles $\theta (\lambda_{i})$ are listed in the respective columns of Table 2. It needs to be further noted that the original thickness of the Dy${}_{2}$O${}_{3}$ sample was only 0.192mm, which caused no issues in the optical transmittance measurements; however, the Faraday rotation measurements with these samples were affected by a large relative error; in particular in the most interesting 2-μm spectral region. The reason was that the induced rotation angle was close to the angular resolution of our setup. Therefore, the original sample was divided into smaller pieces and the obtained slices were sandwiched together in the sample holder. In this arrangement, a larger rotation angle was induced in the Dy${}_{2}$O${}_{3}$ sample, which enabled suppression of the relative measurement error.

### 2.5. Model Functions for the Faraday Rotation Dispersion

As a next step, we have tried to find a satisfactorily accurate models for the obtained data of the Verdet constant dispersion (for the CeF${}_{3}$ and Dy${}_{2}$O${}_{3}$ samples) and of the saturated specific Faraday rotation (for the YIG crystal sample). A general model, which may be used to describe the specific Faraday rotation of a solid-state medium, provided that the described spectral range is sufficiently far from any resonance, may be written as follows [13,47,48,49]
(4)$$\Phi \left(\lambda ,B,T\right)=\sum_{k}\;\left[\frac{E_{k}\left(B,T\right){\lambda_{0,k}}^{3}\left(T\right)\lambda^{2}}{{\left(\lambda^{2}-{\lambda_{0,k}}^{2}\left(T\right)\right)}^{2}}+\frac{F_{k}\left(B,T\right){\lambda_{0,k}}^{2}\left(T\right)}{\lambda^{2}-{\lambda_{0,k}}^{2}\left(T\right)}\right]+G\left(B,T\right).$$

There are four different contributions to the Faraday rotation considered within this approximative model; all of them reflecting the complex phenomena which arise due to the Zeeman splitting of the medium’s energetic states under the influence of the magnetic field. Three of these contributions exhibit a dispersive character and are represented by the two terms in the sum that are proportional to the coefficients $E_{k}$ and $F_{k}$, which are, in general, depending on the medium’s temperature *T* and on the magnetic field *B*. These contributions refer to the so-called diamagnetic (the $E_{k}$-terms), mixing and paramagnetic contributions (jointly represented in the $F_{k}$-terms) and are associated with the allowed electric dipole transitions between the ground and excited states of the medium. The effective transition wavelengths $\lambda_{0,k}$ corresponding to these transitions also vary with temperature. The last term is a wavelength-independent gyromagnetic contribution G(B,T) of the magnetic dipole transitions which have resonances in the far-infrared and microwave regions. A detailed description of all of these contributions as well as their derivation may be found in the available literature [13,47,48,49].

At this point it is worth noting that the character of the wavelength dependence (or independence) of the individual aforementioned contributions to the Faraday rotation remains unchanged regardless of the regime of operation, i.e., the diamagnetic contributions will be always ∝$\lambda^{2}/{\left(\lambda^{2}-{\lambda_{0,k}}^{2}\left(T\right)\right)}^{2}$ and vice versa, but the model coefficients $E_{k}^{\left(l\right)}$, $F_{k}^{\left(l\right)}$ and $G^{\left(l\right)}$ corresponding to the linear regime will differ from those for the saturated regime - $E_{k}^{\left(s\right)}$, $F_{k}^{\left(s\right)}$ and $G^{\left(s\right)}$. In the linear regime, the model coefficients are linearly proportional to the magnetic field, which, considering Equation (Equation 2), may be expressed as
(5)$$\Phi^{\left(l\right)}\left(\lambda ,B,T\right)=V\left(\lambda ,T\right)B=\left\{\sum_{k}\;\left[\frac{E_{k}^{\prime (l)}\left(T\right){\lambda_{0,k}}^{3}\left(T\right)\lambda^{2}}{{\left(\lambda^{2}-{\lambda_{0,k}}^{2}\left(T\right)\right)}^{2}}+\frac{F_{k}^{\prime (l)}\left(T\right){\lambda_{0,k}}^{2}\left(T\right)}{\lambda^{2}-{\lambda_{0,k}}^{2}\left(T\right)}\right]\;+\;G^{\prime (l)}\left(T\right)\right\}B.$$

The individual contributions to the Faraday rotation caused by the Zeeman effect are now represented by the Verdet constant V(λ,T) with the corresponding proportionality factors $E_{k}^{\prime (l)}$, $F_{k}^{\prime (l)}$ and $G^{\prime (l)}$. The model (Equation 5) represents a general model, which we will further simplify for the description of the Verdet constant dispersion of the CeF${}_{3}$ crystal and Dy${}_{2}$O${}_{3}$ ceramics.

In the saturated regime, the specific Faraday rotation is no longer a function of the applied magnetic field, i.e., $\Phi^{\left(s\right)}=\Phi^{\left(s\right)}\left(\lambda ,T\right)$, and the individual Faraday rotation contributions are proportional to the coefficients $E_{k}^{\left(s\right)}\left(M_{s},T\right)$, $F_{k}^{\left(s\right)}\left(M_{s},T\right)$ and $G^{\left(s\right)}\left(M_{s},T\right)$ that are depending on the medium’s saturation magnetization $M_{s}$.

Hereafter, it is assumed that the temperature of the investigated material samples was fixed at the room temperature during the measurement and, hence, the explicit dependence of the model coefficients and the transition wavelengths on temperature will be dropped from now on, bearing in mind that the model parameters will be obtained for a specific fixed temperature $T_{f}$. The explicit dependence of the coefficients $E_{k}^{\left(s\right)}$, $F_{k}^{\left(s\right)}$ and $G^{\left(s\right)}$ on the saturation magnetization will be dropped as well.

#### 2.5.1. CeF${}_{3}$ Crystal

The general model (Equation 5) may be often considerably simplified. For instance, the contribution of the diamagnetic terms is usually very weak compared to the paramagnetic and mixing terms if the described spectral region is well removed from the resonance lines. Therefore, the diamagnetic contribution may be often neglected for the sake of simplicity. Furthermore, assuming that only one dominant transition located at the effective wavelength of $\lambda_{0,1}$ is contributing to the Faraday rotation (together with neglecting the diamagnetic term) will yield the following model for the Verdet constant dispersion
(6)$$V\left(\lambda \right)=\frac{F_{1}^{\prime (l)}\lambda_{0,1}^{2}}{\lambda^{2}-\lambda_{0,1}^{2}}+G^{\prime (l)}.$$

The model (Equation 6) is often termed as the “single-transition model” in the literature and may be conveniently used to describe the Verdet constant dispersion of the CeF${}_{3}$ crystal in our study since the material does not possess any resonance line within the investigated spectral range 0.6–2.3μm (see Section 2.2).

#### 2.5.2. Dy${}_{2}$O${}_{3}$ Ceramics

For the Dy${}_{2}$O${}_{3}$ ceramics, the situation is a bit more complex and the model (Equation 6) cannot be used [40]. The Dy${}^{3+}$ ions possess many resonance lines within the considered spectral region, as was already documented in several spectroscopic investigations of Dy${}^{3+}$-based compounds [50,51,52]. The available spectroscopic data represents an important input into the following considerations. By examining the line strengths of the individual transitions, one may identify the most prominent ones and carefully assess which of them needs to be considered in the model. The transition, which needs to be taken into the account, is located at approximately 1.25μm, corresponding to the ${}^{6}$H${}_{15/2}$→${}^{6}$F${}_{11/2}+$${}^{6}$H${}_{9/2}$ transition. Compared to this transition, the influence of the rest of the transitions located in 0.6–2.3μm region on the Faraday rotation is relatively low and may have only a local impact in the nearest vicinity of the transitions. It may be questionable whether to consider the ${}^{6}$H${}_{15/2}$→${}^{6}$H${}_{11/2}$ transition located at approximately 1.66μm. According to our analysis, the cost of including additional fitting parameters is not counterbalanced by a significant increase in the overall model accuracy and, therefore, we have decided to neglect it in our investigation. The final form of the model function for the Verdet constant dispersion with six fitting parameters may be expressed as follows
(7)$$V\left(\lambda \right)=\frac{F_{1}^{\prime (l)}\lambda_{0,1}^{2}}{\lambda^{2}-\lambda_{0,1}^{2}}+\frac{E_{2}^{\prime (l)}\lambda_{0,2}^{3}\lambda^{2}}{{\left(\lambda^{2}-\lambda_{0,2}^{2}\right)}^{2}}+\frac{F_{2}^{\prime (l)}\lambda_{0,2}^{2}}{\lambda^{2}-\lambda_{0,2}^{2}}+G^{\prime (l)},$$
where the $\lambda_{0,1}$ corresponds to the most dominant transition located in the UV region and $\lambda_{0,2}$ is the effective wavelength of the ${}^{6}$H${}_{15/2}$→${}^{6}$F${}_{11/2}+$${}^{6}$H${}_{9/2}$ transition. The diamagnetic term for the second transition needs to be considered, as its importance grows rapidly in the vicinity of 1.25μm.

#### 2.5.3. YIG Crystal

Description of the Faraday rotation dispersion of the YIG crystal represents a very complex problem and, despite the considerable scientific effort that has been put in, the understanding of its Faraday rotation spectra is still far from complete [13,42,44,53]. The main source of complexity originates in the fact that the Fe${}^{3+}$ ions occupy two different sites in the crystal lattice, the octahedral and the tetrahedral, giving rise to a rather complicated optical and magneto-optical spectra [43,54]. Intensive research in the 1960s has revealed that the octahedral sublattice gives a greater, positive contribution, while the tetrahedral sublattice gives a smaller, negative contribution to the Faraday rotation [55,56,57]. It has been further demonstrated that the dispersive part of the YIG’s Faraday rotation is caused by the electric dipole transitions of the Fe${}^{3+}$ ions located mainly in the UV and the visible part of the spectrum. These contributions may be modelled by the corresponding diamagnetic terms in the model (Equation 4), i.e., the $E_{k}$-terms [56,57]. The follow-up research was focused mainly on deepening the knowledge of the material’s optical absorption spectra as well as on the development of suitable simplifications of the model (Equation 4) that could be used to describe the Faraday rotation induced at the wavelengths attractive for the applications [43,54,58,59,60]. As a result of these investigations, some of the major transitions of the octahedral/tetrahedral Fe${}^{3+}$ ions distributed in the wavelength range of 0.4 to 1.0μm have been identified and their effect on the Faraday rotation in the visible and near-IR has been discussed. Towards the shorter wavelengths, the absorption grows rapidly which is attributed to co-occurrence of the bi-exciton transitions (simultaneous crystal-field transition of two neighbouring ions) and charge-transfer transitions between two Fe${}^{3+}$ ions creating a Fe${}^{2+}$–Fe${}^{4+}$ pair. Nevertheless, to the best of our knowledge, there is still no consensus in the available literature in how exactly these transitions affect the Faraday spectra in the YIG’s transparent window from 1 to 5μm nor which of them are the most influential.

Despite the lack of knowledge, we wish to propose the following set of assumptions, based on which we argue that it is possible to obtain a satisfactorily accurate model for our measured data of the saturated specific Faraday rotation dispersion in the spectral range of 1.1–2.3μm. These assumptions are derived from the conclusions made in the studies of the YIG’s Faraday rotation performed at the near-IR wavelengths and in the spectroscopic characterizations [43,56,58,59,60]. As it follows from these studies, the intense allowed transitions in the UV region arguably represent the major contribution to the dispersive part of the Faraday rotation in the near-IR region. Due to the prevalence of the bi-exciton transitions in the UV, it may be assumed (as a simplified hypothesis) that it is possible to approximate the contributions of these transitions using a single diamagnetic term, in which the corresponding transition wavelength $\lambda_{0,1}$ will represent an effective common wavelength of these transitions and the $E_{1}$ will represent the effective value of $E_{k}$-coefficients of these transitions [58,59,60]. As it further follows, the rest of the transitions located in the 0.4 to 1.0μm region will have a negligible contribution to the Faraday rotation in the near-IR region, except the wavelengths that are close to the 1.0μm. Based on these assumptions, the model (Equation 4) may be reduced to a following simplified form which we will try to use to describe the saturated specific Faraday rotation dispersion of the YIG crystal at the mid-IR wavelengths
(8)$$\Phi^{\left(s\right)}\left(\lambda \right)=\frac{E_{1}^{\left(s\right)}\lambda_{0,1}^{3}\lambda^{2}}{{\left(\lambda^{2}-\lambda_{0,1}^{2}\right)}^{2}}+G^{\left(s\right)}.$$

## 3. Results

In this section, we are going to present the results obtained by the Faraday rotation dispersion measurements along with the errors of the measurement and the fitting parameters of the models discussed in the previous section. All of the presented measurements were performed at a fixed room temperature $T_{f}=22{\;}^{{}^{\circ}}C$.

### 3.1. Dy${}_{2}$O${}_{3}$ Ceramics and CeF${}_{3}$ Crystal

The data obtained for the Dy${}_{2}$O${}_{3}$ ceramics and CeF${}_{3}$ crystal samples are depicted in the Figure 3a together with the relative measurement errors and the relative deviations of the data from the obtained fitted models. It should be noted that, according to the convention, the Verdet constant of Dy${}_{2}$O${}_{3}$ and CeF${}_{3}$ should be negative, because it rotates the plane of polarization of the probe beam in the clockwise sense. We have decided to switch the sign in the plots to make them more convenient. The errors represent the joint uncertainty, taking into the account the uncertainties of the $B_{\mathrm{eff}}$ and *L* values (see Table 2) and the deviations of the gathered intensity data from the cosine-squared function (Equation 1). Up to the wavelength of ∼1.75μm, the relative errors are ∼5% for both Dy${}_{2}$O${}_{3}$ ceramics and CeF${}_{3}$ crystal. As expected, there are some exceptions in the case of Dy${}_{2}$O${}_{3}$ (with higher errors) located in the regions of the considered $\lambda_{0,2}$-transition around 1.25μm and around the transition at ∼1.66μm. In the 2-μm region, the error of the CeF${}_{3}$ starts to steadily grow up to ∼30%, while the error of the Dy${}_{2}$O${}_{3}$ remains <10% till 2.1μm and then grows rapidly. The main reasons for the observed error growth are decreasing values of the rotation angle and a gradual deterioration of the spectral characteristics of the broadband radiation source. With a different radiation source, a stronger magnet, or thicker samples, it would be possible to further decrease the measurement error.

The Figure 3a further includes the fitted model functions. As explained in Section 2.5.2, the multi-transition model (Equation 7) was used for the Dy${}_{2}$O${}_{3}$ ceramics, while for the CeF${}_{3}$ crystal the “single-transition” model (Equation 6) was considered to be sufficient. The fitted parameters with the respective 95%-confidence bounds are listed in Table 3.

At this point, it should be recalled that the models are valid only if we are well removed from the resonance regions. Therefore, the data obtained for the Dy${}_{2}$O${}_{3}$ ceramics around the $\lambda_{0,2}$-transition and the ${}^{6}$H${}_{15/2}$→${}^{6}$H${}_{11/2}$ transition (both indicated with the ‘x’ symbols in the plot) needed to be omitted for the fit. The influence of the rest of the minor resonance lines was suppressed by using a robust least-squares algorithm. We have further omitted the data above 2.2μm, which are affected by a large measurement error. However, given the optical characteristics of the investigated materials, we argue that the obtained models may serve as an approximation up to ∼2.3μm as well. The relative deviation from the fitted models is <15% in the 2-μm region and steadily increases towards 2.2μm.

The values of the Verdet constant obtained by the He-Ne laser measurements were: −329±21rad/Tm for the Dy${}_{2}$O${}_{3}$ ceramics and −119±6rad/Tm for the CeF${}_{3}$ crystal. All of the laser measurements are in a good agreement (<3%) with the broadband measurements.

For the sake of comparison, several relevant values of the Verdet constant were taken from the available literature and included in the two zoomed plots on the right, in the Figure 3b. The present measurement of CeF${}_{3}$ crystal matches very well the previously published data [29,31], which was obtained using different samples and experimental procedures. The only references, to the best of our knowledge, for a pure Dy${}_{2}$O${}_{3}$ Verdet constant comes from the Refs. [40,61]. In the work [61], a −311rad/Tm value at 0.633μm wavelength was reported, which also agrees very well (<3%) with our presented results. Concerning the data from our previous work [40], these exhibit a good agreement in the visible and near-IR spectral regions (see the upper plot of the Figure 3b), but the match is rather poor in the 2-μm region. This is, however, expectable, since the mid-IR Verdet constant values reported in [40] were affected by a large error at the room temperatures, which has prompted our current re-investigation.

### 3.2. YIG Crystal

The data obtained for the dispersion of the specific Faraday rotation of the YIG crystal in the saturated regime is depicted in the Figure 4a; together with the relative measurement errors and deviations from the fitted model under consideration (Equation 8). Please note that since the polarization plane rotation in the saturated regime does not depend on the effective value of the magnetic field (see Equation (Equation 3)), the error is affected only by the uncertainty of the *L* value and the deviations of the intensity data from Equation (Equation 1). The relative error is <10% accross the whole spectral region from 1.1 to 2.3μm and is steadily growing above ∼2μm. The main cause of the increasing error is here, once again, the gradual deterioration of the spectral characteristics of the utilized broadband source. The obtained parameters of the approximative model (Equation 8) are the following: $E_{1}^{\left(s\right)}=57\pm 2\;\left[\frac{10^{3}\mathrm{rad}}{\mu m.m}\right]$, $G^{\left(s\right)}=98.2\pm 0.6\;\left[\frac{\mathrm{rad}}{m}\right]$, $\lambda_{0,1}=0.200\pm 0.008\;\left[\mu m\right]$. The relative deviations of the data from the obtained model (Equation 8) are <5%, except of the wavelengths above 2.2μm, where the error increases. As a next step, we have extracted several values of the specific Faraday rotation from the figures found in the available literature sources [25,44,57,62,63], and included them in the second plot, in the Figure 4b. The deviation of the specific Faraday rotation values found in the available literature from the values given by the model (Equation 8) with the aforementioned fitted parameters is <8% in the <1.8μm, as well as in the extrapolated >3.5μm spectral ranges.

## 4. Discussion

It might be worth discussing the length $L_{45}$ of a magneto-optical element needed for a 45-degree polarization plane rotation (the case of a Faraday isolator) using the results obtained for the investigated materials. For this purpose, we have prepared Table 4, filled with the Verdet constant and specific Faraday rotation values at the emission wavelengths of some of the yet-reported mid-IR laser host materials [1,9]. In the case of the paramagnetic materials Dy${}_{2}$O${}_{3}$ ceramics and CeF${}_{3}$ crystal, let us assume that these would be operated in the linear regime of the Faraday effect, with the effective value of the applied magnetic field equal to 1.6T. Such an assumption is reasonable since permanent magnets with the peak magnetic field values >1.7T were already reported in the past [37,38,39,64] and are still under development; magnets with slightly weaker fields are already commercially available. The assumed value of 1.6T is therefore considered to be a medium value in terms of what is currently technologically feasible. Concerning the YIG crystal, we will assume the saturated regime of operation. The obtained values of the needed $L_{45}$ lengths are listed in the respective fields of Table 4, in the brackets.

Now, let us examine the results in the context of the other materials, which were reported as potential material candidates for the mid-IR Faraday devices. The presented values of the Dy${}_{2}$O${}_{3}$ ceramics Verdet constant are ranging from −27 to −20 rad/Tm in the 2-μm spectral region and, hence, we may conclude that the material exhibit a considerable higher Verdet constant than the state-of-the-art paramagnetic materials mentioned in the introduction (see Table 1). Across the whole 2-μm region, the needed $L_{45}$-length for constructing a 45-degree magneto-optical element ranges from 17 to 24 mm for the Dy${}_{2}$O${}_{3}$ ceramics, which, after the optimization of the fabrication process by varying the sintering conditions and aids, should be feasible with the currently available ceramics fabrication technology. Please note that the Dy${}_{2}$O${}_{3}$ ceramics sample fabricated in the most recent study [46] had 15 mm in length. Further, in case that would be needed, two or more magneto-optical elements may be incorporated in the design of a Faraday device, or a stronger permanent magnet could be used to shorten the needed length of the magneto-optical element. Therefore, we may argue that the Dy${}_{2}$O${}_{3}$ ceramics may be used to construct a mid-IR Faraday device with a paramagnetic material using the currently technologically feasible permanent magnets.

Concerning the CeF${}_{3}$ crystal, its Verdet constant in the 2-μm region requires the $L_{45}$-lengths ranging from 53 to >100 mm. The advantage over the Dy${}_{2}$O${}_{3}$ ceramics and over the other reported materials [33,34,35,36] is that CeF${}_{3}$ crystal is already fabricable in relatively large dimensions [65] and it has been already reported that a high-average-power Faraday device may be realized using this crystal [66]. Therefore, the crystal may be used to construct a Faraday device for the 2-μm lasers, although a relatively strong magnetic field or more than one magneto-optical element will be needed to realize the desired rotation angle.

The results obtained for the YIG crystal suggests that only ∼3–5 mm long magneto-optical elements should be sufficient to induce a 45-degree polarization rotation at ∼2–3 μm wavelengths (provided that the crystal is operated in the saturated regime). Because of the low saturation magnetization of the iron garnets in general, the iron-garnet-based mid-IR Faraday devices may be constructed using relatively weak magnets. This allows to make them more compact, which is of great importance for the applications. The needed length may be further reduced by substituting rare-earth elements to the YIG crystal lattice, which is a known technique to boost the absolute value of the YIG’s specific Faraday rotation in the near-IR region. Nevertheless, some of the commonly implanted rare-earth elements possess strong absorption bands in the mid-IR, making this technique beneficial only for certain mid-IR wavelengths [67]. To the best of our knowledge, there have been no reports examining the influence of substituting the rare-earth elements to the YIG crystal on the material’s performance with a focus on the mid-IR region. Considering the rapidly increasing demand for the mid-IR magneto-active materials, such studies would undoubtedly be extremely beneficial. In the meanwhile, the pure YIG crystal may represent a universal material choice across the whole mid-IR spectrum due to its high transparency in the whole 1–5 μm spectral region.

Finally, although the investigation of the Faraday rotation dispersion is of great importance in the assessment process of magneto-active materials, it is not the only parameter which needs to be considered, when choosing the proper material for a Faraday device. In particular, when the device is designed to withstand high average powers, the final choice should be governed by the values of the magneto-optical figures of merit or the maximum admissible power criteria [16,17], all of which depends also on the thermal conductivity, linear absorption coefficient, thermo-optical, thermo-mechanical and elasto-optical properties of the material. Therefore, one of the next steps in the research of the mid-IR magneto-active materials needs to be investigation of these unknown properties. In particular, reports of the iron garnets’ performance under the irradiation with a high-average-power laser are practically nonexistent, as far as we know.

## 5. Conclusions

The main motivation of this work was to perform a comprehensive investigation of the room temperature magneto-optical properties of both emerging (Dy${}_{2}$O${}_{3}$ ceramics and CeF${}_{3}$ crystal) and established (YIG crystal) magneto-active materials with a focus on the mid-IR region. A broadband radiation source was used in the investigation, which enabled us to study the wavelength dependence of the Faraday rotation in a higher level of detail. The results of the materials’ examination are summarised in the following points.

The Dy${}_{2}$O${}_{3}$ ceramics:has a 1.86–2.3μm transparency window, and, hence, may be used for constructing Faraday devices for the 2-μm lasers. The material is paramagnetic, which implies it may be used in the linear regime of the Faraday effect and tune the desired rotation angle by the strength of the applied magnetic field.exhibit a ∼27–20 rad/Tm (in the absolute value) Verdet constant in the 1.86–2.3μm spectral region. This will allow using the currently commercially available permanent magnets to achieve the standardly needed Faraday rotation angles after the ceramics fabrication process will be optimized.Verdet constant dispersion data in the 2-μm spectral region may be approximated (with a <15% accuracy) by a model (Equation 7) with the parameters listed in Table 3. The model considers contributions of two most dominant electric dipole transitions located at approximately 0.189μm and 1.255μm.

The CeF${}_{3}$ crystal:is transparent up to 2.5 μm and, therefore, it may be also used for the 2-μm Faraday devices. Similarly to Dy${}_{2}$O${}_{3}$, the material is paramagnetic.has a ∼9–5 rad/Tm (in the absolute value) Verdet constant in the 1.86−2.3μm spectral region. Relatively strong magnetic fields or multiple magneto-optical elements would be needed to achieve the standardly needed Faraday rotation angles using the CeF${}_{3}$ crystal, nevertheless, the technology to fabricate this material in large sizes is already available [65].Verdet constant dispersion data in the 2-μm spectral region may be approximated (with a <15% accuracy) by a model (Equation 6), which considers contributions of a single, most dominant electric dipole transition located at approximately 0.234μm. The rest of the model parameters is listed in Table 3.

The YIG crystal:has a wide transparency window from 1 to 5 μm, and, therefore, represents a universal Faraday material choice for the whole mid-IR spectral region. Its ferrimagnetic nature implies the convenience of its operation in the saturated regime in the Faraday devices. Then, the desired rotation angle is tuned by adjusting the length of the magneto-optical element.has a 232–185 rad/m saturated specific Faraday rotation in the 1.86–2.3μm spectral region, which, considering the material’s low saturation magnetization, enables using a relatively weak and compact magnet to achieve the needed Faraday rotation. The material is commercially available in small-scale sizes with a prospect of fabrication of larger-scale samples in the form of transparent ceramics in the near future.Specific Faraday rotation dispersion data in the 2-μm spectral region may be described (with a <10% accuracy) by a proposed model (Equation 8) with the parameters listed in Section 3.2. The model approximates the contributions of the Fe${}^{3+}$ ions’ multiple electric dipole transitions distributed in the UV region using a single diamagnetic term with an effective transition wavelength at 0.200μm. The values given by this simplified model exhibit a <8% agreement with the yet-reported literature values in the <1.8μm, as well as in the extrapolated >3.5μm spectral regions. Although the understanding of the YIG’s UV transitions is not complete, we believe that the proposed model may serve, at least, as a rough approximation of the material’s specific Faraday rotation in the saturated regime over the whole mid-IR spectral region.

## Figures and Tables

**Figure 1 materials-13-05324-f001:**
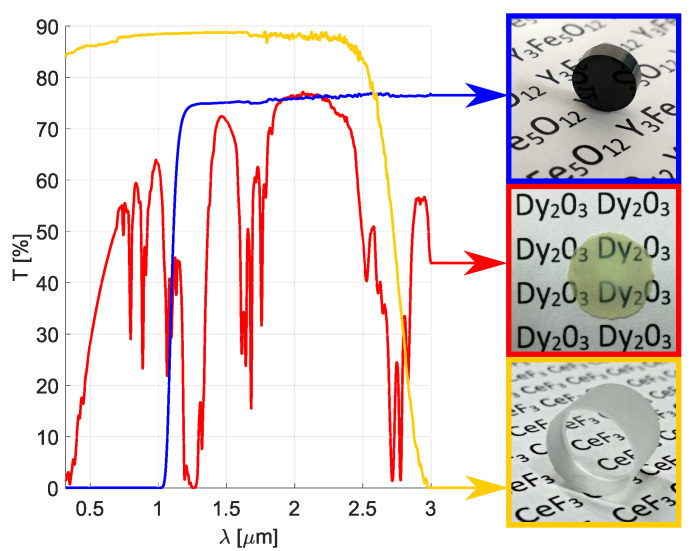
In-line optical transmittance of the investigated samples.

**Figure 2 materials-13-05324-f002:**
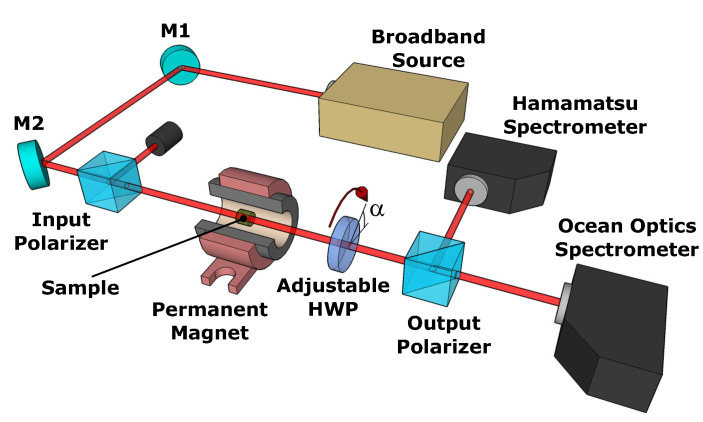
Experimental setup for the Faraday rotation dispersion measurement.

**Figure 3 materials-13-05324-f003:**
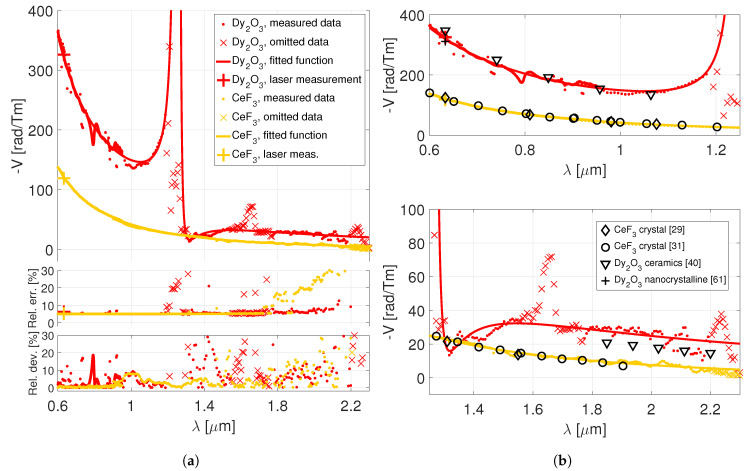
Verdet constant dispersion of the Dy${}_{2}$O${}_{3}$ ceramics and CeF${}_{3}$ crystal. (**a**) The measured data,
omitted data for the fit, fitted models (the top-most plot). Relative errors of the measurement, relative
deviations of the data from the fitted models (the two bottom-most plots). (**b**) Comparison with the
Verdet constant values that may be found in the available literature.

**Figure 4 materials-13-05324-f004:**
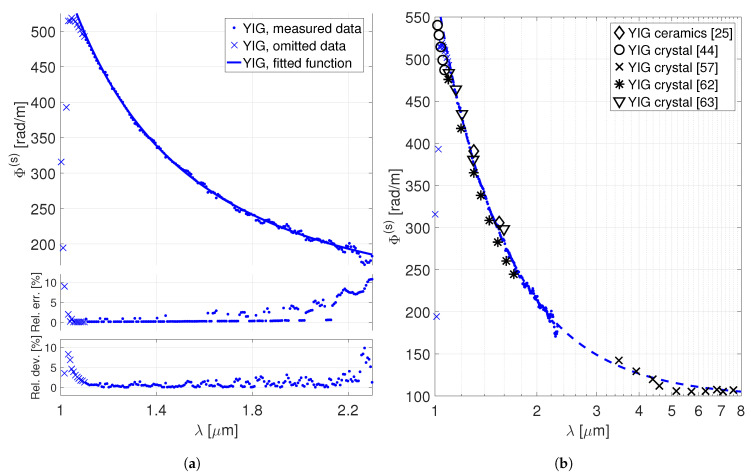
Dispersion of the specific Faraday rotation of the YIG crystal in the saturated regime.
(**a**) The measured data, omitted data for the fit, fitted model (the top-most plot). Relative errors of
the measurement, relative deviations of the data from the fitted model (the two bottom-most plots).
(**b**) Comparison with the values that may be found in the available literature.

**Table 1 materials-13-05324-t001:** The yet-reported paramagnetic or diamagnetic mid-IR magneto-active materials and the absolute values of their Verdet constant at ∼1.94μm [31,33,34,35,36].

Material	$\left|V\right|\;\left[\mathrm{rad}/T.m\right]$ @ ∼1.94μm	Magnetic Behaviour
CeF${}_{3}$ crystal	10.1	paramagnetic
(Dy${}_{X}$Y${}_{0.95-X}$La${}_{0.05}$)${}_{2}$O${}_{3}$ ceramics (X = 0.7–0.9)	10.7–13.8	paramagnetic
EuF${}_{2}$ crystal	12.3	paramagnetic
75 % wt. Dy${}^{3+}$-doped multicomponent glass	7.9	paramagnetic
Te${}_{20}$As${}_{30}$Se${}_{50}$ glass	15.2	diamagnetic

**Table 2 materials-13-05324-t002:** The sample parameters used in the Faraday rotation dispersion calculations. The considered uncertainty of the applied $B_{\mathrm{eff}}$ values is influenced by the magnetic field measurement error and by an undesired displacement of the sample from the maximum of the magnetic field. The estimation of the possible misplacement was purposely overestimated to be up to ±2mm from the maximum. * The value specified for the YIG crystal is listed only for the sake of completeness as the sample was characterized in the saturated regime.

	$L\;\left[\mathrm{mm}\right]$	$B_{\mathrm{eff}}\;\left[T\right]$
Dy${}_{2}$O${}_{3}$ ceramics	0.574±0.001	1.20±0.06
CeF${}_{3}$ crystal	7.553±0.001	1.19±0.06
YIG crystal	2.006±0.001	1.19 ± 0.06 *

**Table 3 materials-13-05324-t003:** Obtained fitting parameters of the Verdet constant dispersion model (Equation 7) (for the Dy${}_{2}$O${}_{3}$ ceramics) and of the model (Equation 6) (for the CeF${}_{3}$ crystal).

	$E_{2}^{\prime (l)}\;\left[\frac{\mathrm{rad}}{\mu m.T.m}\right]$	$F_{1}^{\prime (l)}\;\left[\frac{\mathrm{rad}}{T.m}\right]$	$F_{2}^{\prime (l)}\;\left[\frac{\mathrm{rad}}{T.m}\right]$	$G^{\prime (l)}\;\left[\frac{\mathrm{rad}}{T.m}\right]$	$\lambda_{0,1}\;\left[\mu m\right]$	$\lambda_{0,2}\;\left[\mu m\right]$
Dy${}_{2}$O${}_{3}$ ceramics	−0.40±0.01	−3070±127	10.76±0.05	−3.5±0.2	0.189±0.003	1.255±0.001
CeF${}_{3}$ crystal	-	−789±4	-	3.41±0.02	0.234±0.001	-

**Table 4 materials-13-05324-t004:** Verdet constant and specific Faraday rotation values of the Dy${}_{2}$O${}_{3}$ ceramics, CeF${}_{3}$ and YIG crystals as given by the obtained models at various emission wavelengths corresponding to several yet-reported mid-IR lasers [1,9]. An $L_{45}$ length of a magneto-optical element needed for a 45-degree polarization plane rotation is specified in the brackets for all of the investigated materials. * These values have been extrapolated.

Laser Host Material, Emission Wavelength	Dy${}_{2}$O${}_{3}$ Ceramics, $V\;\left[\mathrm{rad}/\mathrm{Tm}\right]\left(L_{45}\;\left[\mathrm{mm}\right]\right)$	CeF${}_{3}$ Crystal, $V\;\left[\mathrm{rad}/\mathrm{Tm}\right]\left(L_{45}\;\left[\mathrm{mm}\right]\right)$	YIG Crystal, $\Phi^{\left(s\right)}\;\left[\mathrm{rad}/m\right]\left(L_{45}\;\left[\mathrm{mm}\right]\right)$
Tm:silica fiber, 1.86μm	−27.4±2.9(∼17.9)	−9.2±0.1(∼53.2)	232±23(∼3.4)
Tm:germanate fiber, 1.90μm	−26.6±2.8(∼18.4)	−8.7±0.1(∼56.5)	226±22(∼3.5)
Tm:silica fiber, 1.94μm	−25.9±2.6(∼19.0)	−8.2±0.1(∼60.0)	221±21(∼3.6)
Tm:YAG, 2.01μm	−24.6±2.5(∼20.0)	−7.4±0.1(∼66.5)	212±19(∼3.7)
Ho:YLF, 2.05μm	−23.9±2.4(∼20.5)	−7.0±0.1(∼70.5)	208±18(∼3.8)
Ho:YAG, 2.10μm	−23.1±2.3(∼21.2)	−6.5±0.1(∼75.9)	203±17(∼3.9)
Tm:YLF, 2.30μm	−20.1±2.0(∼24.3) *	−4.8±0.1(∼102.2) *	185±15(∼4.2)
Ho:ZBLAN, 2.95μm	–	–	151±9(∼5.2) *

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
