# Peer review of "Faraday Rotation of Dy2O3, CeF3 and Y3Fe5O12 at the Mid-Infrared Wavelengths"

_materials, 2020, doi:10.3390/ma13235324_

Round 1

Reviewer 1 Report

The authors report Faraday rotation of Dy2O3, CeF3 and Y3Fe5O12 at the mid-infrared wavelengths. The content is scientifically interesting ,and organized very well. I suggest the editor accept this manuscript as it is.

Author Response

The authors would like to thank Reviewer #1 for the time he/she spent on reviewing our manuscript and for the positive feedback we have received from him/her. We also appreciate that the reviewer has marked the structure of the document to be well-organized. This is greatly important for the potential readers given the extensive length of the manuscript.

Reviewer 2 Report

    The paper entitled Faraday rotation of Dy2O3, CeF3 and Y3Fe5O12 at the mid-infrared wavelengths by Vojna and coworkers is well written and original. I reccomennd its publication. I have only minor concerns:   - The format of Table 1 should be improved in order to improve its readability, e.g. the first row content should be in bold format. This comment can be applied also to Tables 3 and 4. -The authors claim in the manuscript the possibility of improving Dy2O3 fabbrication process, in this regard thet should at least mention which strategies can be adopted to reach this objective.

Author Response

The authors would like to express their gratitude for the constructive criticism and hard work of Reviewer #2. His/her pieces of advice were included as far as possible and we hope that our revision has improved the paper to a level of his/her satisfaction.

In response to the reviewer's concerns regarding the tables' readability, we have implemented the change suggested by the reviewer. We have set the first rows of content, i. e. the table heads, to be in bold format. We agree that this arrangement contributes to a better readability of the article. 

Concerning the optimization of the fabrication process of the inspected ceramics, it needs to be done by varying the sintering conditions and by including sintering aids. That is a standard procedure that eventually leads to an improved fabrication process and to an enhanced optical quality of the ceramics. This optimization strategy is mentioned in the section dedicated to optical transmittance, nevertheless, given the reviewer's comment, we decided to add an extra explanation also to the discussion part of the manuscript. Both of the text parts addressing the fabrication process optimization are highlighted with red color for easier orientation in the revised manuscript. 

In the response to the ranking of the English language, we have run a more detailed grammar check (UK English version has been used), which eliminated (as we hope) the vast majority of the typos and grammar mistakes we missed in the previous version of the manuscript.

Reviewer 3 Report

This paper reports on a research on the magneto-optical properties of several types of mid-infrared magneto-active materials like Y3Fe5O12 and CeF3 crystals and Dy2O3 ceramics. In order to check the wavelenght dependence, that is one the valuable novelty contribuitions of this nice work, a broadband radiation source with polarization-stepping method was used.
The design of the research was properly set for and indepth investigation. As well the simulations, although it can be improved namely on what concerns the temperature effect on the model functions for Faraday rotation dispersion, were correctely developed and the conclusions are sound and valuable (although the conclusions could be better summarized for sake of clarity).

Author Response

The authors would like to thank Reviewer #3 for the work he/she spent on reviewing our manuscript and for highlighting the key points and benefits of the presented work for the scientific community. This truly provided us with positive feedback in the sense that the fundamental message of the manuscript should be clear to its potential readers.

Regarding the modification of the Faraday rotation model functions to account for the temperature variation, this is done by allowing the model parameters E, F, G to vary with temperature, as it is explained in the corresponding section 2.5 of the manuscript. The variation of the coefficients with temperature is thoroughly described in the literature referenced in this section. In our present investigation, we have performed measurements at room temperature (22.5 °C), at which the temperature variation of the model coefficients is not so rapid as compared with the cryogenic temperatures. 

The authors have reviewed the conclusions' structure and have agreed not to change the way in which the conclusions are presented (in contradiction with the reviewer's concern regarding its clarity). The structure of the conclusions used in the manuscript has been chosen in this way due to the extensive length of the article. We believe that its present form summarizes all of the results in a both compact and comprehensive way. 

And, once again, we have run a more detailed grammar check in the response to the ranking of the English language. We hope that the vast majority of the typos and grammar mistakes were eliminated in the revised manuscript.

Reviewer 4 Report

Dr. D. Vojuna et al. reported Faraday rotation properties of three kinds of materials, Dy2O3, CeF3, and Y3Fe5O12, in the mid-infrared region in their manuscript entitled “Faraday rotation of Dy2O3, CeF3, and Y3Fe5O12 at the mid-infrared wavelengths.” The authors investigated the fabrication process to obtain the materials with sufficient quality for optical measurement. They experimentally measured the Faraday rotation properties with precise manner using a special spectroscopic setup the author made. The spectral behaviors for each material are analyzed with the reasonable approximation models. The results are consistent with the values from literatures in the same wavelength range with the literatures. I consider these results are very important information for development the optics applying magneto-optic effect in mid-infrared region. I have minor comments and questions in some part. 

Comments and questions:

  1. The authors used an achromatic half-wave plate in the experimental setup for the faraday rotation measurement. The achromatic bandwidth was 0.6 – 2.7 μm. How did the authors prepare such a HWP working in so broadband range? Is it commercially available? If so, it is worthy to note the company name and product number of the HWP.

  1. In page 7, the authors mentioned “the contribution of the diamagnetic terms is usually very weak compared to the paramagnetic and mixing terms.” How many orders of magnitude does the contribution of these terms actually differ?

Author Response

The authors would further like to express their gratitude for the constructive criticism and hard work of Reviewer #4. The number-wise answers to the reviewer's specific comments are addressed below. 

  1. Yes, this half-wave plate is actually commercially available at Thorlabs (the part number is SAHWP05M-1700). The working range is, indeed, quite impressive. We have included the part description in the manuscript as well. 
  2. In the case of the Dy2O3 ceramics, the difference in magnitude between the diamagnetic term parameter and the paramagnetic + mixing term parameter (considering the transition at 1.25 μm) is about three orders of magnitude. This actually agrees very well with the yet-performed observations and measurements, as discussed, for instance, in Ref. 13. It needs to be further noted that the actual contributions to the Faraday rotation are further proportional to the λ-2 (param. + mix. term) and λ-4 (diamagnetic term), which makes the contribution of the diamagnetic term even more negligible compared to the paramagnetic term (if the described spectral region is far away from the corresponding transition). In our case, however, the transition lies in the middle of the described spectral range, and the influence of the diamagnetic term needed to be accounted for in the vicinity of the 1.25-μm transition. It has also positively contributed to the accuracy of the model in the mid-IR region. We did not make any changes to the manuscript in response to the reviewer's comment no. 2.

Reviewer 5 Report

Very weak motivation to research. Experimental values of the Verdet constant for these materials are already known for a wavelength of 1.94 μm. The obtained changes in the values of the Verdet constant for the wavelength of 2.3 µm and 1.94 µm fit into the measurement error. Ceramic technology is described in another article [ Vojna, D.; Yasuhara, R.; Furuse, H.; Slezak, O.; Hutchinson, S.; Lucianetti, A.; Mocek, T.; Cech, M. Faraday effect measurements of holmium oxide (Ho2O2) ceramics-based magneto-optical materials. High Power Laser Science and Engineering 2018, 6, e2. doi:10.1017/hpl.2017.37.] and is not the purpose of this article. It is necessary to formulate more deeply the purpose of the study.

There is no information about the samples CeF3 and YIG (are these commercial samples? Provide information about the manufacturer, were there antireflection coatings, etc.)
Figure 1. Does the figure show sample transmission or attenuation?Was Fresnel reflection taken into account?What is the reason for the low transmission of yttrium iron garnet?Is the Dy doped ceramic scattering due to sintering pores or material surface?

What is the originality of your model function for the Faraday rotation variance compared to the model function from the literature cited? What is the reason for the increase in the error for wavelengths above 2.2 μm? By reducing the source power at these wavelengths?

«the original sample was divided into smaller pieces and the obtained slices were sandwiched together in the sample holder.» With this method, you add the loss associated with the reflection from the faces and the thickness increases in proportion to scattering. What kind of transmission was for such a sandwich?

Table 4 is hard to read. maybe you need to change the columns and rows in places?

Author Response

The authors would also like to thank for the critique and hard work of the Reviewer #5. His/her pieces of advice were included as far as possible and we hope that our revision has improved the paper to a level that is now suitable for publication. The number-wise answers to the reviewer's specific comments are addressed below.

1) Reviewer's comment: Very weak motivation to research. Experimental values of the Verdet constant for these materials are already known for a wavelength of 1.94 μm. The obtained changes in the values of the Verdet constant for the wavelength of 2.3 µm and 1.94 µm fit into the measurement error. Ceramic technology is described in another article [ Vojna, D.; Yasuhara, R.; Furuse, H.; Slezak, O.; Hutchinson, S.; Lucianetti, A.; Mocek, T.; Cech, M. Faraday effect measurements of holmium oxide (Ho2O2) ceramics-based magneto-optical materials. High Power Laser Science and Engineering 2018, 6, e2. DOI:10.1017/hpl.2017.37.] and is not the purpose of this article. It is necessary to formulate more deeply the purpose of the study.

1) Our response: The authors have reviewed the paper in detail and have decided not to agree with the reviewer on this point as we are about to explain on the following lines.

The experimental values of the Verdet constant at the 1.94 μm wavelength are, to the best of our knowledge, known only for the CeF3 crystal. In that case, the Verdet constant was reported up to 1.95 μm and considerably affected by a measurement error [31]. For the YIG crystal, we were unable to locate any literature source that would report on the mid-IR saturated Faraday rotation around 2 or 3 μm wavelengths and, concerning the pure Dy2O3 ceramics, the only reported measurement of the Verdet constant was affected by a large measurement error at room temperature [40]. The model of the Verdet constant reported in [40] yields deviated data at near-to-room temperature and, therefore, may not be used for a proper design of a Faraday device. This is explained in the introduction. We have highlighted the specific text parts with red colour for better orientation in the revised manuscript. 

The Dy2O3 ceramics and CeF3 crystal have a transparency window at ~1.84 - 2.4 μm and till 2.5 μm, respectively (see the section on transmittance), in which these materials may be used as Faraday rotator materials, but they have not been fully (or properly) investigated just yet. Knowledge of the Verdet constant at a single wavelength only (e.g. at 1.94 μm) is merely enough for a proper design of a Faraday device since the Verdet constant varies with the wavelength. The same situation applies to the saturated specific Faraday rotation of the YIG crystal. Table 4 clearly demonstrates that the variation with the wavelength needs to be taken seriously: the difference between the Verdet constant/specific Faraday rotation values (as given by the obtained models) at 1.94 and 2.3 μm is statistically significant for all of the investigated materials.  

To summarise, we have investigated the involved materials with a focus on their mid-IR working ranges and have obtained models for the Faraday rotation which describes new, original data to better accuracy than has been reported. That is due to the implementation of a more robust polarization-stepping method. And that is the motivation of this paper. To provide more accurate, original data and discuss the in-depth peculiarities (i. e. which transitions contribute how, etc.) of the materials' mid-IR Faraday rotation. Such a study has not been performed yet.

Supported by the positive feedback from the other four reviewers, the authors feel confident about the structure, motivation, and results presented in this manuscript.

2) Reviewer's comment: There is no information about the samples CeF3 and YIG (are these commercial samples? Provide information about the manufacturer, were there antireflection coatings, etc.) Figure 1. Does the figure show sample transmission or attenuation? Was Fresnel reflection taken into account? What is the reason for the low transmission of yttrium iron garnet? Is the Dy doped ceramic scattering due to sintering pores or material surface?

2) Our response: It would be difficult to implement any anti-reflection coating that would work across the whole measured spectral range of 0.6 to 2.3 μm. We have clarified that no coatings were applied to the samples in the text and specified the manufacturers for the crystal samples. 

Figure 1 presents the transmittance measurement, as specified in the text, as well in the caption of the figure. And yes, this measurement by the spectrophotometer takes into account also the Fresnel losses. The depicted lines represent the fraction of power that is transmitted through the material samples. The origins of the optical losses are the Fresnel losses, scattering, and absorption. In the case of the YIG crystal, the low transmission in the 0.3 to 1.0 μm region is attributed to the existence of the electronic transitions. In the case of the dysprosium ceramics, the origin of the losses would be difficult to analyse from the simple transmittance measurement. Of course, the Fresnel losses could be eliminated by calculation from the refractive index or by comparing the results obtained from two differently long samples, which would allow making a very rough estimation of the total losses induced by scattering and absorption, but this still would not allow separating the scattering and the absorption, nor the origin of the scattering.

Anyway, the main purpose of the transmittance measurement here is to simply demonstrate the possible working ranges of the investigated materials. The core of this investigation is the Faraday rotation analysis. A more sophisticated investigation of the optical losses is done usually in the process of fabrication optimization studies. 

3) Reviewer's comment: What is the originality of your model function for the Faraday rotation variance compared to the model function from the literature cited? What is the reason for the increase in the error for wavelengths above 2.2 μm? By reducing the source power at these wavelengths? 

3) Our response: Our model functions are obtained by simplifications from the general model function for the specific Faraday rotation that is referenced. The simplifications are made by taking into account only the most dominant contributions for each material. This is thoroughly explained in section 2.5. The model functions that we use are not original in the sense that we would derive new models; the originality of this investigation lies in the utilization of a broadband source with the polarization-stepping method, which allows accumulating a larger amount of data (several hundreds of data points across the investigated spectral range). Based on this larger dataset, it is possible to study which contributions are influencing the wavelength dependence of the Faraday rotation in a higher level of detail. Such an investigation would not be possible based on the knowledge of the Faraday rotation on just a few wavelengths.

The increasing error trend above the 2.2 μm is attributed to a gradual deterioration of the broadband source spectral characteristics - the source power gradually diminishes at these wavelengths. Another source of error growth is the decreasing value of the rotation angle. We mention this in the Results section.

4) Reviewer's comment: «the original sample was divided into smaller pieces and the obtained slices were sandwiched together in the sample holder.» With this method, you add the loss associated with the reflection from the faces and the thickness increases in proportion to scattering. What kind of transmission was for such a sandwich?

4) Our response: Yes, this arrangement would increase the reflection losses from the additional surfaces, which is, however, of no concern for the Faraday rotation measurement which is the prime core of our investigation. The main goal is to get a stable, strong enough signal through the setup. As demonstrated in the Results section, the measurement error remained stable till 2.2 μm for the dysprosium ceramics, indicating an excellent following of the cosine-squared function, undisturbed by the multiple reflections.

5) Reviewer's comment: Table 4 is hard to read. maybe you need to change the columns and rows in places?  

5) Our response: We have switched the column-row arrangement for a more convenient representation of our findings.

Round 2

Reviewer 5 Report

Authors' response to comments In that case, the Verdet constant was reported up to 1.95 μm and considerably affected by a measurement error [31]. and, concerning the pure Dy2O3 ceramics, the only reported measurement of the Verdet constant was affected by a large measurement error at room temperature [40].

The article from Ref. 40 [Slezák, O.; Yasuhara, R.; Vojna, D.; Furuse, H.; Lucianetti, A.; Mocek, T. Temperature-wavelength dependence of  Verdet  constant  of  Dy2O3 ceramics. Optical Materials  Express  2019, 9, 2971–2981. doi:10.1364/OME.9.002971] was made by practically the same team; does this work present data for the same ceramics or for another? If for another, how do they differ? That is, it looks as if in this work you are citing some of the data that is in work [40], only determined with a smaller error and a slightly larger spectral range. In article 40 and this one, even the same pictures of the experimental setup

Authors' response to comments “For the YIG crystal, we were unable to locate any literature source that would report on the mid-IR saturated Faraday rotation around 2 or 3 μm wavelengths”

A quick review of the literature provides the following information on the Verdet constant in the ~ 2 micron range.

For YIG

  1. Stevens, T. Legg and P. Shardlow, “Integrated disruptive components for μm fibre lasers (ISLA): project overview and passive component development,” Proc. SPIE, 9730 973001 (2016). http://dx.doi.org/10.1117/12.2207613 PSISDG 0277-786X

Specific rotation angle of polarization for a wavelength of 2.1 μm from the article you are citing [57]. Krinchik, G.S.; Chetkin, M.V. TRANSPARENT FERROMAGNETS. Soviet Physics Uspekhi 1969, 12, 307–319. doi:10.1070/pu1969v012n03abeh003902.

And the experimental values of the Verdet constant in cerium fluoride, according to your statement, are already known

Taking into account the data given by me, what is the novelty and originality?

 Authors' responseYes, this arrangement would increase the reflection losses from the additional surfaces, which is, however, of no concern for the Faraday rotation measurement which is the prime core of our investigation. The main goal is to get a stable, strong enough signal through the setup. As demonstrated in the Results section, the measurement error remained stable till 2.2 μm for the dysprosium ceramics, indicating an excellent following of the cosine-squared function, undisturbed by the multiple reflections.

I probably phrased my question badly. My question was the following, knowing the refractive index of the material, you can calculate the reflection loss from the interface. The refractive index of dysprosium oxide is ~ 1.9, which gives a reflection from each surface of the order of 10%. that is, on each plate you lose 20% of its reflection and about 5% for scattering. Does this affect the measurement accuracy, taking into account that you say in the article that the main cause of measurement errors is the low signal intensity in the area of more than 2.2 μm

Author Response

The authors would also like to express their gratitude for the additional round reviews, namely to Reviewer #5. The number-wise answers to the additional concerns of the reviewer are addressed below.

1) Reviewer's comment: The article from Ref. 40 [Slezák, O.; Yasuhara, R.; Vojna, D.; Furuse, H.; Lucianetti, A.; Mocek, T. Temperature-wavelength dependence of  Verdet constant of Dy2O3 ceramics. Optical Materials  Express  2019, 9, 2971–2981. doi:10.1364/OME.9.002971] was made by practically the same team; does this work present data for the same ceramics or for another? If for another, how do they differ? That is, it looks as if in this work you are citing some of the data that is in work [40], only determined with a smaller error and a slightly larger spectral range. In article 40 and this one, even the same pictures of the experimental setup. 

1) Our response: The measurement that was reported in the Ref. 40 differs from the present measurement in several ways. Different sample, different setup, different aim of the research.

First and foremost, the sample that has been used in the Ref. 40 was different, although it was also a pure Dy2O3 ceramics. The sample that was characterized in our previous work was a bit shorter and had a worse optical quality (residual porosity etc.). The sample that was characterized in the present manuscript was manufactured as a part of a different batch, created after we published the former paper. The motivation for the re-inspection was to obtain more accurate results in the mid-IR window of the material since the data published in 2019 has been affected by a large error (>~80 %) in this window at room temperatures. The main problem causing such a large error was that the induced rotation angle was close to the resolution limit of our setup. The sample that has been characterized in the present investigation has been set up from smaller pieces of a NEW sample with better optical quality. In this manner, we have been able to increase the rotation angle induced in the mid-IR and, hence, determine the Verdet constant with higher accuracy. In the former paper, we have indicated a possibility of using the Dy2O3 ceramics for the mid-IR wavelengths, but it is the present study, in which we were able to demonstrate the potential of this material's utilization for the Faraday devices with the currently available permanent magnets. That is important for the application point of view.

Secondly, the former measurement from the year 2019 was temperature-dependent, i. e. the sample was kept in a cryostat enabling temperature control. So, although the method is basically the same, i. e. the polarization-stepping technique, the experimental setup is different. In the present investigation, the sample was fixed in a special optomechanical holder, enabling sandwiching the ceramics pieces together. This is a major difference between the schematics of the experimental setups as well. While in the former publication a cryostat is depicted, in the current manuscript there is only a sample with a mention in the text that it is fixed in a special holder. The rest of the setup is, however, the same, since the method that has been used is the same. 

Thirdly, the major aim of the former publication was different. The main goal was to demonstrate the need for considering the multi-transition (three trans. were considered) model to describe the Verdet constant as a function of wavelength and temperature. In other words, the scope was more general, mainly material-oriented. The present manuscript is more oriented on the mid-IR wavelengths since this was a drawback of the previous investigation. The focus is more on the determination of the potential for the applications, although it is not on the expense of the theoretical background of the currently presented model, which now considers two major transitions. The theoretical background is more extensive in the present manuscript. 

We hope that this statement has addressed this concern of Reviewer #5 regarding the originality of the data, as well as it will answer a potential concern of any of the future readers, which is enabled by the open review process of this journal.

2) Reviewer's comment: A quick review of the literature provides the following information on the Verdet constant in the ~ 2 micron range.

For YIG

  1. Stevens, T. Legg and P. Shardlow, “Integrated disruptive components for μm fibre lasers (ISLA): project overview and passive component development,” Proc. SPIE, 9730 973001 (2016). http://dx.doi.org/10.1117/12.2207613 PSISDG 0277-786X

Specific rotation angle of polarization for a wavelength of 2.1 μm from the article you are citing [57]. Krinchik, G.S.; Chetkin, M.V. TRANSPARENT FERROMAGNETS. Soviet Physics Uspekhi 1969, 12, 307–319. doi:10.1070/pu1969v012n03abeh003902.

2) Our response:  We are familiar with the proceeding paper that the reviewer suggests, but, given the fact that the paper does not provide any information about the measurement procedure that has been used to obtain the presented Verdet constant data, or about the magnetic field that was used, nor it mentions the parameters or dimensions of the samples, we do not consider this proceeding as a very reliable source of information. Also, it needs to be remarked, that the YIG crystal is, due to its ferrimagnetic nature, would be usually used in the saturated regime in the Faraday devices. The slope of the growth of the rotation angle vs the applied field in the linear regime (reported in the proceeding) is for a ferrimagnetic material depending also on the demagnetizing factors, which are given by the shape of the sample. This is why the quantity of saturated specific Faraday rotation is used in the literature to compare these materials (as explained in section 2.4). And, as far as we know, there have been no reports of this quantity for YIG in its mid-IR window, apart from the referenced measurement from 1969 [57].

To address the second part of the comment, the measurement on the ~2.1 μm wavelength from the reference [57] has been performed with a rather spectrally broad source of radiation (2.1 +- 0.4 μm is specified), which is why the presented values may be regarded only as a "mean" value for the ~0.8 μm wide variety of wavelengths. Considering the magnetic field we have applied, the value agrees to a <8 % accuracy, similarly to the other values presented in that paper for the wavelengths above 3.5 μm. So, even if we would consider this single data point, there is still an unknown space around the 3 μm wavelengths that have not been reported.

Anyway, it also needs to be noted that we do not present the measurement of the YIG crystal as a "cutting-edge" new, as we are aware that the material has been studied for decades now. The material serves here mainly as reference material, although measurement of the specific Faraday rotation on the ~2 - 3 μm wavelengths using a broadband source has not been reported yet. What is interesting about this material, however, is that there seems to be no consensus on the model for the mid-IR Faraday rotation of this material. In section 2.5.3, we briefly review the current state of knowledge and based on that we propose a simplified model for the specific Faraday rotation that gives rather well literature-matched predictions. This is, as we believe, the main contribution of this work regarding the YIG crystal: a model that will allow making a rough estimation of the Faraday rotation on the yet-unmeasured mid-IR wavelengths.     

3) Reviewer's comment: And the experimental values of the Verdet constant in cerium fluoride, according to your statement, are already known.

3) Our response: We are not aware of making any statement about the CeF3's Verdet constant to be known. We have stated in our last response, the Verdet constant of this material is known only up to 1.95 μm. In the present investigation, we have been able to measure the VC up to 2.3 μm for the cerium fluoride. And, as we have discussed in the previous round of reviews, the difference between the values obtained for these wavelengths is statistically significant and needs to be taken seriously when designing a Faraday device. 

4) Reviewer's comment: Taking into account the data given by me, what is the novelty and originality?

4) Our response: This question has been to some extent already answered in the responses above, which is why we will only summarise, underlining the main benefits of this work.

The Verdet constant of the dysprosium sesquioxide ceramics in its whole mid-IR window (1.86 - 2.3 μm) has been determined with a considerable better accuracy, which revealed that the material may be used with the presently available permanent magnets. This revelation is of great importance for the applications.

The Verdet constant of cerium fluoride has been measured for 1.95 - 2.3 μm wavelengths for the first time.

The saturated specific Faraday rotation of YIG crystal has been measured for 1.8 - 2.3 μm wavelengths for the first time using a broadband radiation source. Based on the new data, a model for the Faraday rotation of this material has been derived and may be conveniently used to estimate the Faraday rotation in the unknown 3-μm spectral region.

The manuscript presents new data on the Faraday rotation of all of the involved materials using an original polarization-stepping measurement method. Utilization of the broadband source and the mentioned technique has enabled to discuss the origins of the Faraday rotation of these materials in a higher level of detail as compared with the common single-wavelength measurements.

A benefit is also that these current material options for constructing a 2-μm and possibly 3-μm Faraday devices are presented in this manuscript alongside. 

Considering all that, we believe that the paper brings multiple benefits to potential readers from the relevant field and that the novelty and originality of this work are considerable. And, again, given the other four reviews that were very positive, the authors firmly believe that the present version of the manuscript is scientifically correct, properly motivated and appealing to the field it contributes to.  

The key information from this reply is already included in the relevant parts of the manuscript. Nevertheless, we have made a few minor changes to the text to ensure minimization of arising misunderstandings. For easier orientation, we have, once again, highlighted this information in the revised version.